# Evaluation of Retained Austenite in Carburized Bearing Steel Using Magneto-Inductive Method

**Laura G. Ionescu** [1,2], **Mangesh V. Pantawane** [3], **Constantin Tănase** [2], **Răducu V. Sichim** [2], **Catalina A. Dascălu** [1] **and Brândușa Ghiban** [1,*]

1  Metallic Materials Science and Physical Metallurgy Department, University Politehnica of Bucharest, 313 Splaiul Independentei, 060042 Bucharest, Romania; laura.ionescu@timken.com (L.G.I.); catalina.dascalu@hotmail.com (C.A.D.)
2  Materials Science Research and Development, Timken S.A, 100525 Ploiesti, Romania; tanasc.tanase@timken.com (C.T.); raducu.sichim@timken.com (R.V.S.)
3  Materials Science Research and Development, Timken World Headquarters, North Canton, OH 44720, USA; mangesh.pantawane@timken.com
*  Correspondence: ghibanbrandusa@yahoo.com

**Abstract:** The present work explores the magneto-inductive method to evaluate different levels of retained austenite content in carburized 20NiCrMo7 bearing steel while comparing the corresponding measurements by X-ray diffractometry and image analysis by optical microscope. The content of retained austenite in carburized 20NiCrMo7 steel was modified with additional tempering to yield three samples with distinct ranges of retained austenite profiles in the carburized region of the steel. The retained austenite measured at different depths in these samples using the magneto-inductive method had an outcome comparable to other methods. Further discussion based on data suggests that the magneto–induction method yields precise (with an average deviation of 0.5%) results with sufficient sensitivity at different levels (including below 5 vol. %.) of retained austenite.

**Keywords:** retained austenite; bearing steel; 20NiCrMo7 steel; microstructure; non-destructive testing; XRD; magneto–induction method





## 1. Introduction

Retained austenite (RA), as the name suggests, is the austenite phase retained in the steel after its incomplete transformation to martensite/bainite/ferrite. The presence of RA in steel can induce positive or negative changes, depending on its fraction. Hence, an appropriate range of RA content is desired in finished products such as gears, bearings, and tools. For making bearings more durable in various applications, their microstructure must be appropriately tuned as per their use. For example, bearings in gearboxes can experience premature failure due to the development of cracks in the bearing steel if the required RA content is not maintained [1–3]. It was reported that increased RA content (>20%) in these bearings provides a relatively long service life in such applications [4–6].

The transformation of RA to martensite or other phases during service can affect the performance of the steel [7–10]. In general, high RA content negatively affects wear resistance and dimensional stability and increases the risk of abrasive cracks. On the other hand, low RA can impact fracture toughness and rolling contact fatigue life. Transformation of RA in service causes dimensional instability of hardened bearing components. Hence, RA can be detrimental for the application requiring tighter tolerance, while more RA can be beneficial in the scenario mentioned earlier [5,11–15]. Consequently, optimizing RA content to achieve the best combination of desired properties is important.

Based on the importance of RA mentioned above, optimizing RA by process and composition and accurately measuring that content is imperative. Current techniques to measure RA include X-ray diffraction (XRD), neutron diffraction, scanning electron

microscopy (SEM), electron backscattered diffraction (EBSD), optical microscopy (OM), Mössbauer spectroscopy, dilatometry, and techniques utilizing the magnetic properties of phases in steel [16–18]. As per the different principles and probing volumes associated with each of these techniques, their accuracy and precision change. Of all the above-mentioned techniques, XRD and OM are the most frequently utilized techniques to quantify RA both in industry and research institutions [19–21]. However, these techniques also carry limitations due to their principles of measurement. XRD involves measuring the diffracted peak positions and intensity of diffracted X-rays that are particular to a specific crystalline phase and its volume fraction [7,21,22]. RA measurement by analysis of an optical micrograph is done based on the contrast of the image, which is obtained by etching the polished sample [23]. However, both techniques are destructive and time-consuming. Furthermore, due to the nature of these techniques, they cannot be effectively utilized to inspect 100% of industrial parts [7,9,10,22]. Considering the limitations posed by XRD due to its longer scanning time, their optimized versions are commercially available and easy to use. The analysis time in this instrument is reduced by selectively scanning the samples in the $2\theta$ range, where the desired phases are expected to constructively interfere. However, since the peaks are collected selectively, carbide correction is not possible. In addition, the measurement time is still in the range of 10–20 min. The Mössbauer spectrometer is another commercially available mobile tool that is optimized for measuring RA. However, the measurement still requires more than 20 min at a given location. Moreover, the instrument has limited mobility due to attached auxiliary devices to analyze the signals [24].

Techniques utilizing the magnetic properties of steel are, thus, often an attractive alternative to measuring paramagnetic constituents such as RA. The magneto–inductive method is one of those techniques that primarily measures the ferromagnetic content present in steel. Feritscope® is a commercially available device that works on the magneto–inductive principle. The device has two coils integrated into it, one of which produces pulses of magnetic field that permeate through the steel samples up to several millimeter depths. The ferromagnetic content, such as ferrite/martensite/bainite, responds to the change in magnetic field by generating an electric field in the second coil. The change in electric field proportional to the ferromagnetic content present in the steel is measured by the second coil. Austenite, being a paramagnetic phase, can be measured using primarily ferromagnetic phases [25–27]. The measurement of RA austenite using Feritscope® is completed in a couple of seconds. The magnetic measurement is a time-efficient, simple, and non-destructive method for quantitative measurement of RA [18,26].

The magneto–inductive method has been extensively explored to measure ferromagnetic content in steel welds [25,26,28–31]. Very few reports have attempted to focus on comparing this technique with other methods to detect RA content [14,18,27,30–34]. Since the magnetic field generated by the Feritscope® penetrates several millimeters deeper, it is considered to provide bulk quantification. To the best of the author's knowledge, the capability of Feritscope® to detect the varying RA content in the case carburized region for a given probing volume has still not been explored [35].

In view of the above-presented scenario, the present study explores the magneto-inductive method to measure RA content in carburized bearing steel. In an effort to validate this technique, it has also been compared with other commonly used methods to measure RA, such as XRD and OM. In addition, the sensitivity of the Feritscope® device to measure different levels of RA has been studied.

## 2. Materials and Methods

### 2.1. Material

This study used carburized roller samples made of 20NiCrMo7 steel received from The Timken Company. The length and diameter of the bearing roller are 85 mm and 48 mm, respectively, as depicted in Figure 1.

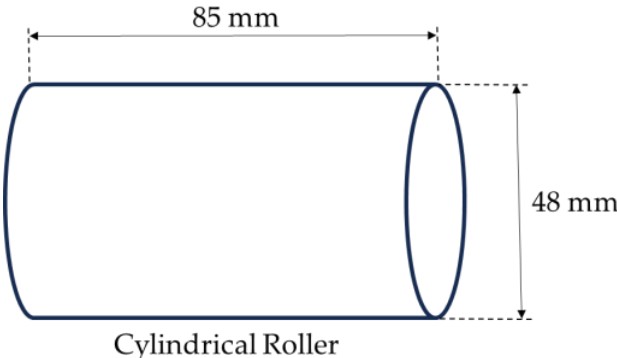

**Figure 1.** Schematic of a cylindrical bearing roller with its dimensions.

The general flow of the carburizing cycle used for the roller sample is presented in Figure 2a. All samples of the same dimensions and uniform composition going through the same thermokinetics of the carburizing cycle were assumed to have similar RA content (Figure 2a). In addition to the carburizing cycle, a few samples were exposed to two hours of secondary tempering at 260 °C and 290 °C in order to reduce RA content (Figure 2b). These tempering temperatures were selected based on the studies reporting the effect of tempering temperatures on RA content [23,25,26]. Essentially, three distinct roller samples were considered for evaluation with specific tempering stages, as presented in Table 1. It should be noted that primary tempering is a part of the carburizing cycle used for all the samples (Figure 2a).

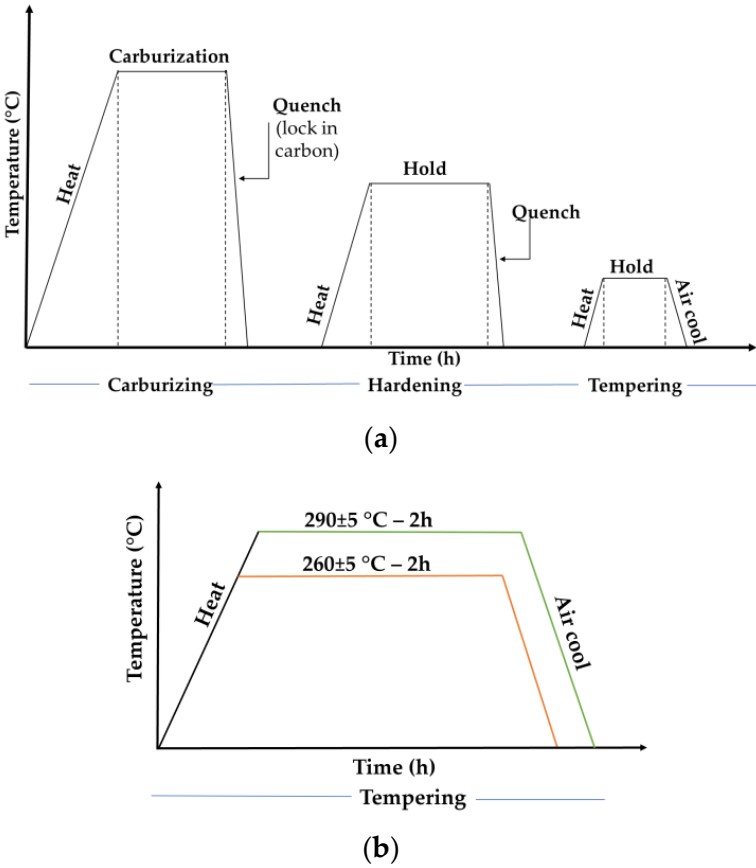

**Figure 2.** Schematic of heat treatment process showing (**a**) carburizing cycle; (**b**) secondary tempering treatment.

**Table 1.** Combination of tempering stages exposed to the samples.

| Sample # | Tempering Temperatures |
|---|---|
| 1 | Primary tempering: 182 °C ± 5 °C |
| 2 | Primary tempering: 182 ± 5 °C + Secondary tempering: 260 ± 5 °C |
| 3 | Primary tempering: 182 ± 5 °C + Secondary tempering: 290 ± 5 °C |

All the samples were evaluated using optical emission spectrometry (OES) to check the composition of the samples as per the standard BS EN ISO 683-17. In addition, a hardness depth profile was obtained using the Vickers microhardness tester EMCO-TEST at a load of 1 kgf to measure the carburized case depth of the samples.

*2.2. RA Measurement*

2.2.1. X-ray Diffraction

All the samples were evaluated using X-ray Diffraction (XRD) in accordance with ASTM E915 and ASTM E975-13. The measurements were performed on a Proto LXRD® diffractometer with a chromium α X-ray source (λ = 2.291 Å) and a vanadium metal filter. The X-ray scan was performed on an instrument using a 20.48-degree-wide PSSD detector consisting of 512 channels. Due to the instrument configuration, the scan is performed in discrete steps to quickly capture the RA content. The complete 20.48-degree wide-scan is collected from all channels simultaneously. Part of the samples were electropolished at different depths to avoid incorporating stress that may otherwise be induced during mechanical removal of material, such as grinding and polishing. The electropolishing process was carried out using an electrolyte containing distilled water, ethanol, perchloric acid, and butoxyethanol at 22 °C and a voltage of 20 V. These parameters were run for 3 min to remove 0.025 mm of material.

2.2.2. Optical Microscopy

All surfaces of the specimens to be used as visual evaluation samples were cut, mounted, milled, and polished according to standard procedures. For the determination of the RA, the samples were etched using Nital 4% to obtain sufficiently high contrasts. The RA content was measured using Stream Essentials Olympus image analysis software. The same procedure was followed at different depths at the same location where RA was evaluated using XRD after electropolishing.

2.2.3. Magneto-Induction Method

The martensite content was measured in the center of the sample body by the magneto–induction method using a Fischer Feritscope® FMP30. Since it is a non-destructive technique, the five measurements were collected on the cleaned surface of the samples without any special surface preparation. The measured martensite content by the axial probe of the Feritscope® deviates from the actual (true) content if the surface has a significant curvature. In the present case, since the roller sample had a curved surface due to its diameter of 48 mm (Figure 1), the correction factor suggested by Fischer Feritscope® was used to obtain the true ferromagnetic content as given in Equation (1).

$$Fe_T = Fe_m * k \tag{1}$$

where $Fe_T$ is the actual (true) ferromagnetic content in the material, $Fe_m$ is the measured ferromagnetic content by Feritscope®, and $k$ is the correction factor for a given curvature [34]. Martensite, being ferromagnetic, is essentially measured using the magneto–induction method by Feritscope®. Eventually, using the actual martensite content ($M = Fe_T$), the RA was obtained as per Equation (2)

$$RA \% = 100 \% - Fe_T \% \tag{2}$$

However, the correction factor was only considered at the surface and not at different depths. This is because the samples were electropolished at different depths and were sufficiently flat to not consider the curvature effect.

In the present work, carbide content was assumed to be insignificant enough to not affect the measurement of *RA* content in the carburized region by the above techniques. In order to compare the outcome of each method mentioned above, the same locations were probed by each method by electropolishing at different depths, such as 0.2 mm, 0.5 mm, 1.0 mm, and 1.5 mm. All the techniques eventually yield a percentage of *RA* in vol. %; thus, any reference to *RA* % should be considered as vol. % throughout the article.

### 3. Results and Discussion

*3.1. Chemical and Mechanical Properties*

The chemical composition of the roller was measured using optical emission spectrometry, as presented in Table 2. It was confirmed that the sample was made from 20NiCrMo7 according to BS EN ISO 683-17 [36].

**Table 2.** Chemical composition (in wt%) of 20NiCrMo7 measured by optical emission spectrometry and as per BS EN ISO 683-17 standard.

| Element | C | Mn | Si | Cr | Ni | Mo |
|---|---|---|---|---|---|---|
| Measured | 0.23 | 0.69 | 0.25 | 0.64 | 1.61 | 0.25 |
| Standard | 0.17–0.23 | 0.4–0.7 | 0.15–0.35 | 0.35–0.65 | 1.6–2.0 | 0.2–0.3 |

The microhardness profile of all the samples is shown in Figure 3. The effective Case Hardening Depth (CHD) is generally measured as the depth below the surface where the Vickers' hardness drops to 550 HV. The microhardness traversal was made from the surface to a depth of 4.5 mm. The effective CHD measurements obtained from the hardness profile are presented in Table 3. While all the samples were subjected to the same carburizing treatment, the effect of secondary tempering on the CHD of samples #2 and #3 is noticeable. This, in turn, is also indicative of varying RA as a function of the secondary tempering temperature exposed to samples #2 and #3.

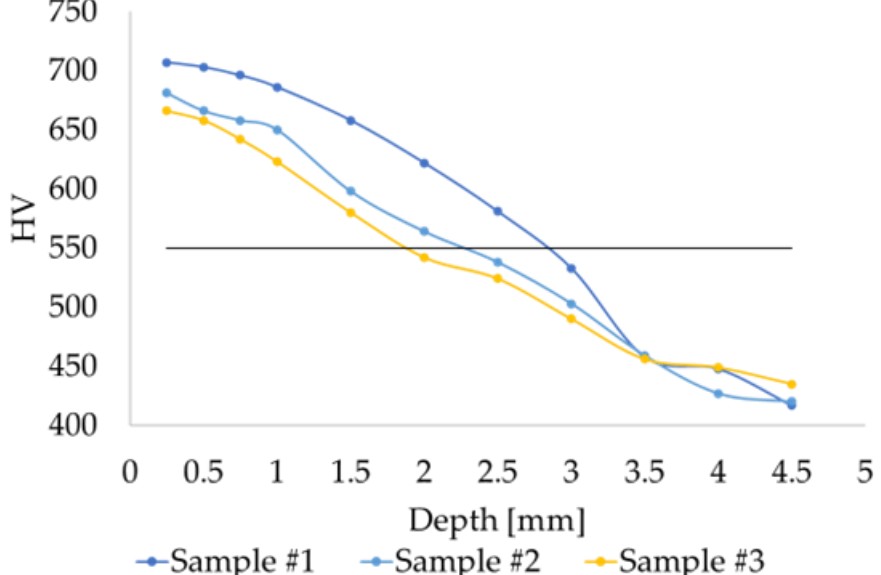

**Figure 3.** Effective CHD measurement by microhardness.

**Table 3.** Effective CHD measurement by microhardness.

| Sample # | CHD [mm] |
|----------|----------|
| Sample #1 | 2.8 |
| Sample #2 | 2.3 |
| Sample #3 | 1.9 |

### 3.2. RA Measurement by X-ray Diffraction

Figure 4 shows representative XRD peaks of martensite and austenite obtained for each sample. It should be noted that the XRD scan was conducted in discrete ranges only to cover the peaks corresponding to martensite and austenite. Under the same operating conditions, the relative change in the peak intensities of martensite and austenite can be noticed among different samples.

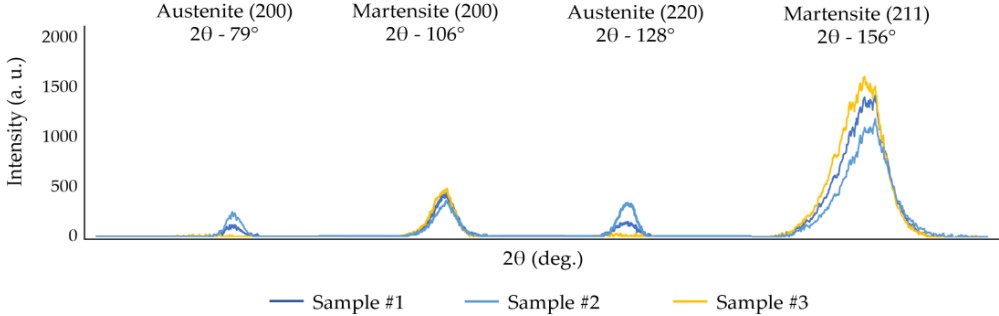

**Figure 4.** XRD peaks corresponding to austenite and martensite for samples #1, #2, and #3.

The RA content in vol. % was measured based on the relative peak intensities of martensite and austenite at different depths of each sample. These results are presented in Table 4. It should be noted that the uncertainty of measured RA is associated with the standard ASTM E975-13 [21].

**Table 4.** RA (in vol. %) at different depths measured by XRD.

| Sample # / Depth | Surface | 0.2 mm | 0.5 mm | 1 mm | 1.5 mm |
|----------|---------|--------|--------|------|--------|
| Sample #1 | 25.9 ± 1.9 | 30.7 ± 2.8 | 30.9 ± 2.7 | 21.7 ± 1.6 | 17.9 ± 1.1 |
| Sample #2 | 11.4 ± 1.1 | 10.1 ± 1.0 | 8.9 ± 0.9 | 5.1 ± 0.9 | 3.8 ± 0.3 |
| Sample #3 | 1.1 ± 0.1 | 2.1 ± 0.3 | 2.7 ± 0.4 | 1.1 ± 0.1 | <1.0 |

Different evolutionary trends in the percentage of RA can be observed depending on the tempering temperature and depth (Table 4).

In sample #1, the average RA content increased from 25.9% to 30.9% up to 0.5 mm and decreased consistently to 17.9% at a depth of 1.5 mm. On the other hand, in the samples tempered at higher temperatures (samples #2 and #3), overall RA decreased compared to sample #1. For sample #2 (retempered at 260 °C), the average RA content appeared to steadily drop from 11.41% at the surface to 3.8% at 1.5 mm, with a percentage decrease of 66.6%. For sample #3 (retempered at 290 °C), no trend was noted in the RA percentage. The values of RA appeared to fluctuate from the surface to 1.5 mm in a range of 1.08% to 2.7%. However, a slight decrease in average RA values from the depth of 0.5 mm to 1.5 mm can be noticed in sample #3.

### 3.3. RA Measurement by Optical Microscopy

Exact locations probed by XRD were parallel investigated using optical microscopy. Figure 5a–c, showing the representative microstructure of all samples, confirmed the presence of martensite along with austenite and insignificant carbides in each sample. In

addition, the representative OM grayscale images of each sample are shown in Figure 5d–f. The appearance of homogeneous martensite (dark) and RA (white) in grayscale allows easier measurement of the individual phase fraction. The effect of varying combinations of primary and secondary tempering treatments in each sample is reflected in grayscale images with varying fractions of RA phase (white). An evaluation made based on the grayscale images to obtain areal % of RA is presented in Table 5. Assuming the random distribution of grains and isotropic grain growth, areal percentage was considered equal to vol. %. OM-based measurement of retained austenite content at different depths had a similar trend as measured by XRD. However, with this method, it is difficult to accurately quantify a low RA < 3%.

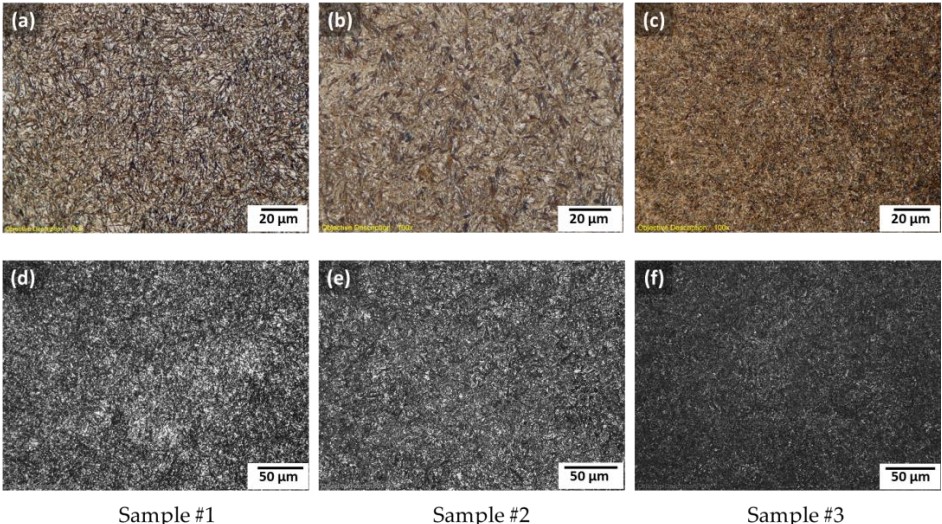

**Figure 5.** Representative optical micrographs showing presence of martensite and RA at 1000X in sample #1 (**a**), sample #2 (**b**), and sample #3 (**c**); and (**d–f**) grayscale images of sample #1, #2, and #3, respectively, at 500X.

**Table 5.** RA (in vol. %) measured at different depths by OM.

| Sample #   Depth | Surface | 0.2 mm | 0.5 mm | 1 mm | 1.5 mm |
|------------------|---------|--------|--------|------|--------|
| Sample #1        | 27      | 28.9   | 28.4   | 20.8 | 12.7   |
| Sample #2        | 11.6    | 11     | 9.4    | 4.1  | <3     |
| Sample #3        | 3.5     | 4.1    | 5      | <3   | <3     |

### 3.4. RA Measurement by Magneto-Induction Method

Figure 6 shows the comparison of the measured ($RA_m$) and actual/true ($RA_T$) RA (in vol. %) at the surface of the samples considered in the present work. As mentioned in Section 2.2, the difference between $RA_m$ and $RA_T$ arises due to the convex curvature of the roller, which leads to non-uniform spacing between the probe and the surface of the sample. The average actual/true RA value appeared to be less than the measured RA value within the sensitivity limits of Feritscope®.

The true retained austenite ($RA_T$) values measured by the magneto–induction method at the surface is close to the values obtained by X-ray diffraction and image analysis (Figure 6). As mentioned earlier, the correction factor was not considered at a depth other than the surface as the curvature effect is insignificant after electropolishing. Thus, instrumentally measured values of RA at different depths for all samples are presented in Table 6. It can be noticed that the values of RA measured at different depths are comparable to those obtained by XRD and optical microscopy. In addition, the insignificant deviation of the RA values measured by Feritscope® indicates a precise measurement of RA compared

to XRD. It appears that Feritscope® is adequately sensitive to detect the RA below 5% with values compared to those obtained by XRD.

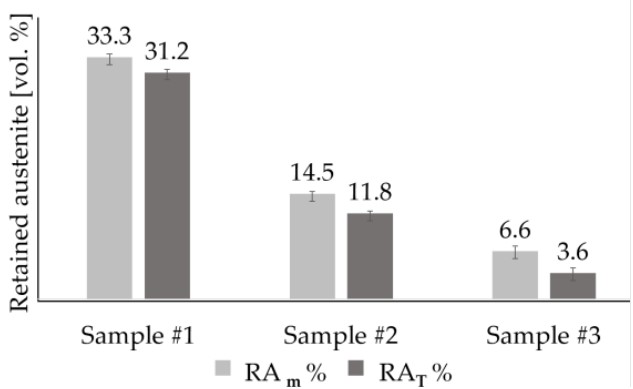

**Figure 6.** Measured and actual content of RA (in vol. %) by magneto-induction method.

**Table 6.** RA (in vol. %) measured at different depths by magneto-induction method.

| Sample #    Depth | Surface | 0.2 mm | 0.5 mm | 1 mm | 1.5 mm |
|---|---|---|---|---|---|
| Sample #1 | 31.2 ± 0.4 | 29.8 ± 0.5 | 29.1 ± 0.7 | 23.9 ± 0.4 | 21.5 ± 0.4 |
| Sample #2 | 11.8 ± 0.4 | 11.5 ± 0.5 | 10.1 ± 0.5 | 9.7 ± 0.5 | 6.2 ± 0.5 |
| Sample #3 | 3.6 ± 0.8 | 3.4 ± 0.6 | 4.4 ± 0.7 | 2.9 ± 0.5 | 1.5 ± 0.4 |

### 3.5. Discussion

In order to compare the results provided by each method, the measured RA values are summarized for easier visual comparison in Figure 7. It should be noted that for samples #2 and #3, the immeasurable low RA values measured by optical microscope and XRD have not been included in the plot. This suggests the inadequate sensitivity of these methods for low RA content. On the other hand, measurement via the magneto–induction method is possible at different levels of RA. In addition, Feritscope® is an easy and quick tool to get consistent results that are in good agreement with those obtained by XRD and optical image analysis. The following discussion focuses on the comparison of each technique with other techniques in detail.

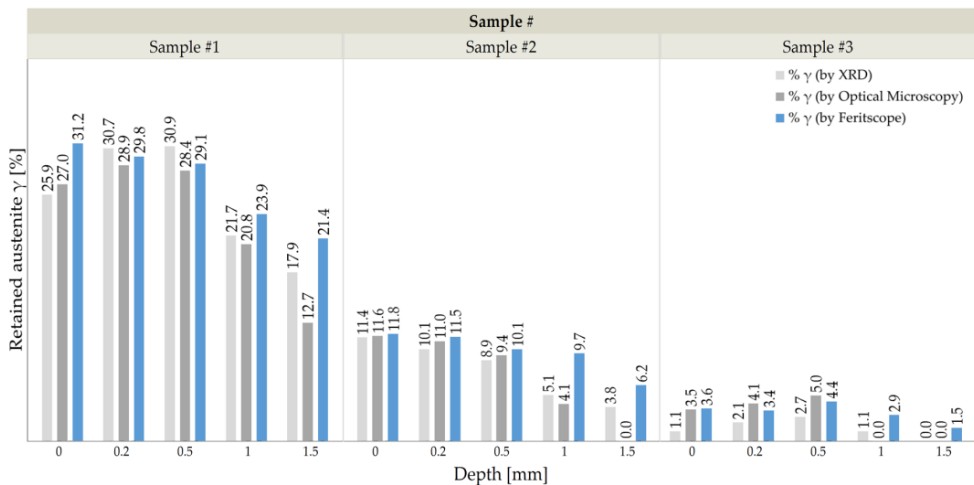

**Figure 7.** Comparison of RA values measured by all three methods: Feritscope®, XRD, and quantitative image analysis using optical micrographs.

The probing region for both XRD and OM can be considered to be nearly the same. This is because the OM yield results from the surface of the sample, and the X-ray penetrates several micrometers, thus being limited to the subsurface region. The values presented in Table 7 were calculated as percentage differences in RA content among the different methods used in the current study. The percentage difference factor was particularly used to provide a comparison factor other than the difference between the absolute values of RA. The graphical presentation of average percentage differences to compare these techniques is shown in Figure 8.

**Table 7.** Summarizes measured data showing percentage differences between methods: X-ray diffraction, Optical Microscopy, and Feritscope®.

| Method | Depth Sample # | 0 mm | 0.2 mm | 0.5 mm | 1 mm | 1.5 mm | Average |
|---|---|---|---|---|---|---|---|
| XRD—OM | Sample #1 | 4% | 6% | 8% | 4% | 33% | 11% |
| | Sample #2 | 2% | 8% | 5% | 21% | <23% | <12% |
| | Sample #3 | 69% | 49% | 46% | <63% | <66% | <59% |
| XRD—Feritscope® | Sample #1 | 18% | 3% | 6% | 10% | 17% | 11% |
| | Sample #2 | 3% | 13% | 13% | 62% | 49% | 28% |
| | Sample #3 | 106% | 47% | 47% | 90% | 40% | 66% |
| OM—Feritscope® | Sample #1 | 14% | 3% | 2% | 13% | 51% | 16% |
| | Sample #2 | 2% | 4% | 7% | 81% | <70% | <33% |
| | Sample #3 | 3% | 18% | 12% | <3% | <66% | <42% |

Samples #1 and #2 had high (20–30%) and medium (4–12%) RA contents, respectively, as measured by XRD and OM. The average percentage difference for these samples ranged between 11 and 12%. The sample #3 with RA content less than 4% is difficult to read with high accuracy under an optical microscope. A small difference in values (1% or 2%) comparable to original values up to a maximum of 5% can lead to high percentage differences, causing the comparison method chosen to provide an apparently high discrepancy in results for sample #3. Moreover, at lower RA fractions, the comparable carbide content may start interfering with measurements by both XRD and OM. While carbide correction could be done in both cases, it is highly challenging to measure the carbide fraction using both techniques. For RA in the range of 5–31 vol. % with insignificant carbide content, it is reasonable to conclude that the results of both methods are within an acceptable range of difference in measurement. XRD and OM being the current most frequently used techniques to measure RA content, it is reasonable to consider them to validate the results obtained by Feritscope®.

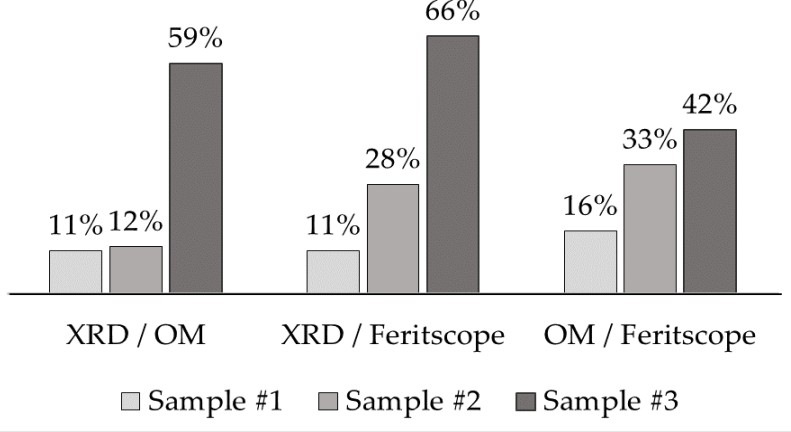

**Figure 8.** Comparison of average percentage differences of RA amongst three methods: Feritscope®, XRD, and image analysis on optical micrographs.

For sample #1 with high RA content (20–30%), Feritscope® measurement had an average percentage difference of 11% and 16% at different depths compared to XRD and

OM, respectively. With these values, it is reasonable to consider that the results of both methods are within an acceptable range of difference in measurement.

For sample #2 with medium RA (4–12%), the average percentage difference of measurement by Feritscope compared to XRD and OM appears to increase to 28–33%. The careful observation suggests that the average percentage difference increases for the lower RA values measured at deeper depths (1–1.5 mm). Similarly, for sample #3 with low RA (<5%) content, the average percentage difference remains high in the range of 42–66%. These increased values at lower RA content point to the same mathematical explanation provided earlier. However, the absolute difference between the measured RA values by these techniques still remains low, making it reasonable to accept the outcome.

As mentioned earlier, the penetration depth of XRD is limited to several micrometers, whereas the measurement by magneto-induction method allows to capture data from several millimeters deeper isotropic volumes. The penetration depth of Feritscope® as per several reports ranges between 1 and 2 mm [32,34]. Thus, it is expected to measure the average RA content present in the probing volume. Considering this scenario, if the surface RA measured by Feritscope® for sample #1 (31.2%) is compared with the average values of RA measured by XRD up to 1 mm depth (27.3%), it seems that the Feritscope® may have slightly less than 1 mm probing depth. The surface RA value measured by Feritscope® for sample #1 is close to the average RA content measured by XRD up to 0.5 mm depth (29.1%). This indicates the penetration depth of Feritscope® is close to 0.5 mm. It also directs us to another hypothetical scenario, where the Feritscope® could be measuring ferromagnetic content with the maximum possible resistance from paramagnetic content, such as RA for deeper penetration > 1 mm. In other words, Feritscope® could be measuring the lowest ferrite content present in the probing volume, which is present at the surface and increases towards the core of the sample. As a result, the RA measured by the Feritscope® at the surface is comparable to the values measured by XRD. Nevertheless, these outcomes warrant additional detailed studies specifically dedicated to evaluating the penetration depth of Feritscope® with varying RA content. Regardless of the measuring mechanism, Feritscope® was found to detect different levels of RA content comparable to XRD and OM. In addition, readings from Feritscope® had slightly higher precision than XRD.

While OM and XRD were considered as references to validate the Feritscope® outcome, it is worth noting the factors that might affect the accuracy and efficiency of these techniques. OM with image contrast counting techniques can produce reliable phase content results. However, necessitating the destructive preparation of multiple mounts while following a lengthy metallographic process and quantifying multiple regions makes this technique highly time-inefficient. The quantification of the RA and the martensite phases with this technique is often difficult due to the very small carbide sizes within the microstructure. Although the carbide fraction is considerably less, it adds to the actual RA austenite fraction as carbides also appear bright in the microstructure (using nital), thereby affecting the measured RA value by OM. In addition, since the measurement is limited to area fraction by OM, anisotropically grown grains might provide deviated results from the actual content.

Quantitative X-ray diffraction techniques (XRD) are considered more accurate techniques than OM for RA quantification. The results are more reliable with XRD, as it can overcome the resolution difficulties of OM. Standard XRD techniques (such as internal standard and area peak fitting) have limitations for complex multiphase materials. While carbide correction could be done, it is highly challenging to measure carbide fractions with other techniques. Additional problems affecting accuracy include preferential orientation and overlapping carbide peaks. The XRD technique could be considered quasi-destructive when sample preparation is required for depth profiling of RA and while evaluating complex-shaped components. Furthermore, both XRD and OM require trained staff to operate.

Feritscope® is relatively easy to operate with quick output, and any complex shaped steel component with a finished surface can be tested. Careful considerations are required while testing the component, such as maintaining the required spacing between probe and sample and considering the correction factor for curved surfaces to get results closer to

the actual content. While the carbide fraction is considered insignificant enough to affect the results, it may still affect them when its fraction is comparable to retained austenite. Interestingly, several studies have reported that iron carbides are ferromagnetic [37,38]. Since the major fraction of carbides in steel is iron carbide, it is likely to join the ferromagnetic response by martensite to the secondary coil of Feritscope®. As a result, the RA measured by Feritscope® is likely to be closer to the actual content by avoiding carbide interference. However, it is worth noting that carbides other than iron carbides, such as chromium carbides, could be paramagnetic and may hinder the final outcome of the magneto–inductive method [37,39]. However, the steel grade considered in the present study contains sufficiently low Cr and Mo content to not form a significant amount of carbide to affect the outcome of magneto–inductive measurements, and the fraction of iron carbide is expected to be higher than the rest of the carbides.

### 4. Conclusions

The results of RA content in carburized bearing steel measured by the magneto–inductive method compared with the results obtained by XRD and OM are summarized below:

- The results of Feritscope® measurements on the sample surfaces revealed a direct correlation with both destructive methods, such as XRD and OM;
- The average percentage difference of RA measured by Feritscope® with XRD and OM was in the acceptable range for RA content in the range of 5–31%;
- The average percentage difference of RA measured by Feritscope® when compared to XRD and OM increased with the reduction in actual RA content, while the absolute difference between the measured RA values remained consistent;
- Feritscope® detected RA content below 5%, and the provided results were comparable to the values measured by XRD and OM;
- Benchmarking Feritscope® against the widely used XRD and OM techniques, it was observed that Feritscope®, with faster detection and less time consumption, provides comparable results at different levels of RA satisfactory enough for industrial purposes.

**Author Contributions:** Conceptualization, L.G.I.; reviewing and editing, M.V.P.; methodology, C.A.D., L.G.I. and C.T.; software, R.V.S.; validation, C.T., R.V.S. and B.G.; formal analysis, L.G.I.; investigations, L.G.I.; data curation, C.T.; writing (original draft preparation), L.G.I.; writing (review and editing), B.G. All authors have read and agreed to the published version of the manuscript.

**Funding:** This work has been funded by the European Social Fund from the Sectoral Operational Programme Human Capital 2014–2020, through the Financial Agreement with the title "Training of PhD students and postdoctoral researchers in order to acquire applied research skills—SMART", contract no. 13530/16.06.2022—SMIS code: 153734.

**Data Availability Statement:** Not applicable.

**Conflicts of Interest:** The authors declare no conflict of interest.

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
