# Peer review of "Evaluation of Retained Austenite in Carburized Bearing Steel Using Magneto-Inductive Method"

_crystals, doi:10.3390/cryst13081173_

Round 1
Reviewer 1 Report
Dear authors,
1. Can you mention the precision of RA determination by new method in the abstract?
2. Page 1, Line 45: correct to “Mössbauer”.
3. Page 2, lines 50-56: for discussion, there are also commercially available industry XRD instruments, mobile, easy to use, also some Mössbauer spectrometers are presented; all optimized for RA analysis.
4. Page 2, line 65: In whole text where, non-ferromagnetic austenite (phases) is discussed, which is not detected by new method. There are also some paramagnetic iron carbides (not cementite as ferromagnetic), which are presented also in bearing steels (like grade 100Cr6, i.e.), see also ref. Suwalski, J.; Kucharski, Z.; Lukasiak, M. Determination of retained austenite in bearing steel. Hyperfine Interact. 1986, 29, 1491–1494. Here “low-temperature carbides” are identified as paramagnetic component in Mössbauer spectrum.
5. Table 1, please can you present the precision of temperature control in the teble, like in Figure 2?
6. Page 4, line 137: can you mention from which depth, thickness is RA information gained? Something is mention on page 2 line 62, and in the discussion. But I feel it will be better to describe it in the experimental method description. My opinion.
7. Page 5, Figure 3 has very low quality, please improve it and increase its size.
8. Page 5, RA measurement by XRD – for discussion – this measurement in selected interval of angles, for specific diffraction peaks – does it confirm or deny the occurrence of iron carbides?
9. Page 6, table 4. Dot at the end of caption. Also, I propose to unify the accuracy of measurement values and errors to the same number of decimal places. The same in table 6.
10. Page 6, line 215, in continuation with my previous discussion – OM did not prove the presence of carbides, also XRD? Results of magneto-inductive method can be affected by the presence of various paramagnetic iron carbides, do you agree?
11. Page 8, discussion, I appreciate the discussion of the amount of information from different layer thicknesses for different materials. I agree that in the case of the MI method, it is an average value from a much wider layer than in other methods and in addition much higher compared to the sampling interval. As also discussed on page 9 lines 289 and 309.
12. Page 10, line 348 – reference number is missing. And again, in some cases, iron carbides can be paramagnetic and may not be detected as a response of Feritscope (see suggested reference). I wonder if the authors have experience with this in bearing steels generally.
13. References 3, 4, 28 are not fully cited.
14. What is the situation, when RA is in an amorphous state (nanometric layer, i.e.), can this affect the analysis by MI method? Maybe just for internal discussion, not in the paper.
Thank you for answers.
Reviewer 2 Report
Abstract, line 12 change "20NiCrMo" for "20NiCrMo7"
Figure 1 is not necessary. Or for example, authors can replace it by a picture of the actual sample.
Line 99 change "sample" for "samples"
Figure 3 must be large
Line 179 replace 3# for #3
What is the precision in the measurement of RA?
Line 207 replace 1.5 % for 1.5 mm
Line 223, replace "retain" for "retained"
Lines 294-295 Which technique is the most frequently used? In this phrase, this is not clear
Line 314: it is not clear why authors deduce that penetration of Feritoscope is 1 mm
Line 318 change "was found detect" for "was found to detect"
Lines 320-321 The two resulted phrases must be joined
Line 348 complete the reference xx
References 8, 9 12 and 13 are not correctly written.

English grammar and syntax should be checked, ideally by a native speaker.
